# Food Composition Impacts the Accuracy of Wearable Devices When Estimating Energy Intake from Energy-Dense Food

**DOI:** 10.3390/nu11051170

**Published:** 2019-05-24

**Authors:** Giulia Lorenzoni, Daniele Bottigliengo, Danila Azzolina, Dario Gregori

**Affiliations:** Unit of Biostatistics, Epidemiology and Public Health, Department of Cardiac, Thoracic, Vascular Sciences and Public Health, University of Padova, 35131 Padova, Italy; giulia.lorenzoni@unipd.it (G.L.); daniele.bottigliengo@studenti.unipd.it (D.B.); danila.azzolina@unipd.it (D.A.)

**Keywords:** energy intake, energy-dense food, automatic food intake measurement

## Abstract

The present study aimed to assess the feasibility and reliability of an a3utomatic food intake measurement device in estimating energy intake from energy-dense foods. Eighteen volunteers aged 20–36 years were recruited from the University of Padova. The device used in the present study was the Bite Counter (Bite Technologies, Pendleton, USA). The rationale of the device is that the wrist movements occurring in the act of bringing food to the mouth present unique patterns that are recognized and recorded by the Bite Counter. Subjects were asked to wear the Bite Counter on the wrist of the dominant hand, to turn the device on before the first bite and to turn it off once he or she finished his or her meal. The accuracy of caloric intake was significantly different among the methods used. In addition, the device’s accuracy in estimating energy intake varied according to the type and amount of macronutrients present, and the difference was independent of the number of bites recorded. Further research is needed to overcome the current limitations of wearable devices in estimating caloric intake, which is not independent of the food being eaten.

## 1. Introduction

Eating habits are well known to play a key role in affecting the onset and the clinical course of noncommunicable diseases [1]. Promoting population health through a healthy diet might represent one of the primary goals of public health policies worldwide [2].

The adoption of a healthy diet requires a detailed assessment of the population’s actual dietary habits [3], allowing for the identification of unhealthy dietary patterns (e.g., excess caloric intake) and their subsequent modification in favor of healthier habits. Dietary assessment is relevant for both primary and secondary prevention of noncommunicable diseases. First, dietary assessment is directed at healthy subjects to maintain their health status (primary prevention of noncommunicable diseases). Dietary assessment is even more relevant to monitor compliance with recommended dietary patterns and prevent complications in subjects already affected by noncommunicable diseases.

However, dietary assessment represents a matter of concern. Several tools are available (food frequency questionnaire, food diaries, 24-h dietary recall) to measure food intake, and they are widely used in clinical research. Unfortunately, such traditional methods present several limitations, such as underreporting (due to recall and desirability biases), poor precision in reporting portions size, and details of food preparation [4]. Therefore, it has been shown that self-reported caloric intake might be inaccurate. In recent years, there has been growing interest in the development of technological tools to objectively assess food intake, overcoming the limitations of traditional dietary assessment methods. Several tools have been developed to date, by different modes of action [5]. Some tools consist of the recording of chewing and swallowing movements [6] using an ad hoc microphone (e.g., AutoDietary [7]) or a piezoelectric sensor (Piezoelectric sensor-based necklace with accelerometer [8]). Such tools look similar to a necklace (incorporating the device, allowing for the recording of swallowing). Briefly, data stored are sent to a smartphone for food recognition through ad hoc algorithms. Other tools estimate food intake through the recording of food pictures (e.g., e-Button [9], “Digital Photography of Food Method” (DPFM) [10,11,12,13], and wearable microcameras [14]). However, such methods have been suggested to have some limitations, e.g., the need for a mobile device for recording food pictures [15]. A user-friendly tool is the Bite Counter (Bite Technologies, Pendleton, USA) [16]; it looks similar to a watch, but has an integrated accelerometer and gyroscope, and it records wrist movements while eating.

Undoubtedly, such tools are promising and provide encouraging results, but they still present several limitations. Some of the tools are not meant for daily use (they are uncomfortable and require training to be used). Not least, most of the tools have been validated only on a small set of food types, and their validity seems to vary according to food type [16]. Therefore, the aim of the present study was to assess the feasibility and reliability of automatic food intake measurement via a bite counting (AFIM-BC) device in estimating energy intake from energy-dense foods and how eventual estimation errors are dependent on the specific food composition.

## 2. Materials and Methods

### 2.1. Participants

Healthy volunteers were recruited at the University of Padova using recruiting flyers. Inclusion criteria were age over 18 years, no allergies to food served during the eating session, no cognitive impairment, and the ability to read and speak in Italian. Before study participation, the volunteers were asked to sign informed consent forms.

According to the Italian legislation framework, approval from the Institutional Review Board was not needed.

Before the eating session, participants underwent anthropometric assessment. The anthropometric assessment was performed by registered dieticians. Weight, height, hip and waist circumference were measured. Body mass index (BMI) and waist-to-hip ratio (WHR) were calculated.

### 2.2. Study Device

The device used in the present study was the Bite Counter (Bite Technologies, Pendleton, USA), a tool that looks similar to a watch (and that operates as a digital watch when it is not counting bites). It must be worn on the wrist of the dominant hand, since it records wrist movements while eating. The rationale of the Bite Counter is that the wrist movements that occur in the act of bringing food to the mouth (using hands or other utensils) present unique patterns that are recognized and recorded by the Bite Counter through the gyroscope.

The application software of the Bite Counter was employed to download and save the data (number of bites) of each eating session (by connecting each device to the computer in which the Bite Counter software was installed).

### 2.3. Study Procedures

Subjects were observed in a McDonald’s restaurant in the Veneto region (Italy). The McDonald’s restaurant was chosen because it allowed for the consumption of standardized meals in terms of meal size, calorie content, and nutrient content. Each subject chose the food items he or she preferred among those available. The nutritional facts of each food item chosen were available on the Italian McDonald’s website (https://www.mcdonalds.it/) and are reported in Table 1.

All subjects finished their meal, and the nutritional facts of the meal of each subject are reported in Table 2.

The eating session of each subject-one at a time- was supervised by trained researchers. Each subject was asked to wear the Bite Counter on the wrist of the dominant hand, to turn the device on before the first bite and to turn it off once they finished the meal. Each subject was told to eat always with the dominant hand and to drink with the other hand (beverages were excluded from the analysis to limit the variability). All participants adhered to the instructions for using the Bite Counter. They all ate using their hands, except the three subjects who had a salad (who ate it using a fork).

### 2.4. Power Analysis

The study was designed to estimate the association between bites and the different nutrients composing the food administered to study participants. A linear regression model was assumed to describe this association and an effect size of 0.5 was assumed as the target minimal association to be detected. With an alpha level of 0.05 and a power of 0.80, a two-tailed test based on the F distribution (non-centrality parameter of 3) indicates that 18 subjects are needed to be enrolled. Computations have been made using G-Power [17] software.

### 2.5. Statistical Analysis

Descriptive statistics were reported as medians (I and III quartiles) for continuous variables and percentages (absolute number) for categorical variables.

Energy intake, based on the number of bites recorded by the Bite Counter, was estimated using three different approaches described by Salley J. (2013) [18], Scisco et al. (2014) [19], and Salley et al. (2016) [20]. Salley J. (2013) [18] provided two different equations (one for males and one for females) to estimate the amount of kcal per bite. Such an equation is based on the subject’s age, weight, and height. Scisco et al. [19] considered that a bite corresponds to 11 kcal for females and to 17 kcal for males (which is similar to the amount of kcal per bite resulting from the application of the equations presented by Salley J (2013)). Salley et al. (2016) [20] provided an equation based on age, sex, and anthropometric measures (height, weight, and WHR) to estimate the number of kcal per bite.

The agreement between the subject’s actual caloric intake, based on the nutritional information reported in Table 2, and the estimated caloric intake by each approach was assessed using a Bland-Altman plot [21]. The association of the content of seven nutritional values with the error in estimated caloric intake, defined as the McDonald’s nutritional information minus the estimated caloric intake based on Bite counter, was evaluated across the approaches mentioned above to understand how the accuracy of estimated energy intake may vary relative to different nutritional content values. In addition, we tested the correlation between the number of bites and nutritional content values. Statistical analysis was performed using the R software system (version 3.5.1) [22].

## 3. Results

Of the eighteen subjects enrolled, 11 were women. The median age was 28.5 years. Regarding anthropometric measures, subjects presented a median BMI of 21.72 and a median WHR of 0.80. None of the subjects were obese. The median actual caloric intake and nutritional values, with I and III quartiles, are reported in Table 3.

The accuracy of caloric intakes were significantly different among the methods used, with the Salley (2016) approach showing the lowest median error (Salley (2013) 229.27 95% C.I. 109.97, 367.19; Scisco (2014) 242.00 95% C.I. 53.75, 355.75; and Salley 2016 56.81 95% C.I. −179.16, 183.43; *p*-value = 0.01). Associations between the error in estimated energy intake and content in (g) of nutritional values for each approach are reported in Table 4. The rows where the author’s names of the proposed approaches are presented contain the percentiles of errors in the estimated caloric intake in Kcal, computed as the difference between the McDonald’s nutritional information and the estimated caloric intake.

Regarding the Salley (2013) approach, carbohydrate and fat content levels were significantly different across categories of errors in estimated caloric intake defined by quartiles (*p*-value 0.030 and 0.017, respectively), whereas only fat content was significantly different for the Scisco (2014) approach (*p*-value 0.027). Regarding the Salley (2016) method, only carbohydrate and dietary fiber content levels were significantly different across errors of estimated energy intake (*p*-value 0.040 and 0.032, respectively). Figure 1 shows the Bland-Altman plots used to evaluate the agreement between the actual caloric intake and the estimated caloric intake by each approach. Overall, Salley (2013) and Scisco (2014) methods showed similar agreement patterns and they overestimate the caloric intake of nearly 200 Kcal, on average. Salley (2016) method showed larger variability in the differences (95% C.I. –470.12, 456.50) than Salley (2013) (95% C.I. –161.82, 584.21) and Scisco (2014) (95% C.I. –178.85, 579.18). Figure 2 reports the relationship between the nutritional content values and error in estimated caloric intake by the different approaches.

Salley (2016) systematically underestimates the caloric intake for lower levels of nutritional value, while Salley (2013) and Scisco (2014) systematically overestimate caloric intake. The overestimation increases at higher nutritional content levels, except for protein and sodium. Table 5 reports the association between the number of bites and the macronutrient content, measured as an increase in (g) for one more bite with relative 95% Confidence Interval (C.I.) and *p*-value.

No significant association was found, as suggested by 95% C.I. that always include the 0 and P-values always greater than 0.05. This pattern is also shown graphically in Figure 3.

## 4. Discussion

The present results showed that the accuracy of the Bite Counter in estimating energy intake varied according to the type and amount (especially for fat and carbohydrates) of macronutrients present and that this variability was independent of the number of bites recorded. Such findings are consistent with previous studies in the field. Wearable dietary assessment devices have been found to be accurate in the counting of bites in both controlled and uncontrolled settings [16,23] and to outperform energy estimation compared to that of self-reporting [20]. However, consistent with the results of the present study, Scisco et al. [19] showed that, even if significant, the correlation with actual energy intake is only moderate. In addition, it has been suggested that the validity of the Bite Counter (in counting the actual number of bites) is different according to food category and that the device is more accurate when subjects eat using their hands [16].

The present findings suggest that such wearable devices, although promising, still present several limitations. The main limiting factor is the variability in the accuracy of energy intake estimation according to the macronutrient content, especially if the device is employed to estimate energy intake from energy-dense foods. Such a limitation could depend on the type of utensils used, which have been shown to influence the wrist movements (e.g., the number of bites required to eat soup with a spoon is often underestimated because the movement of the wrist is reduced to not spill the soup [16]) and on the variability in the size of bite. However, it is worth pointing out that such a study was conducted in a controlled setting to minimize the interindividual variability in meal consumption. Subjects ate the meal under the supervision of the researcher, all ate using their hands (except for the three subjects who ate the salad) and were properly trained on the use of the device. Furthermore, the number of bites was found to be not associated with the macronutrient content. Therefore, such variability in estimating energy intake could be more likely related to the lack of a reliable and feasible algorithm to identify the energy intake from each bite.

AFIM-BC devices might be considered a promising opportunity for monitoring energy intake in the primary and secondary prevention of diet-related noncommunicable diseases. Therefore, the fact that the accuracy of the device in estimating energy intake is lower for high Kcal macronutrients, especially for fats and carbohydrates, is critical because it implies a poor performance of such devices in estimating energy intake from energy-dense foods. Measuring energy intake from such food items is of great interest. It has been widely demonstrated that the increasing availability of inexpensive, energy-dense food has led to an increase in obesity rates. Subjects who eat such types of food items are at risk of excessive daily caloric intake that together with sedentary behaviors, may result in weight gain [24,25]. As an example, subjects enrolled in the present study had a mean energy intake of 718 kcal, adding the energy (168 kcal) of a medium-sized carbonated beverage (which was excluded from the analysis), resulted in an energy intake of 886 kcal from a single meal, which represents almost half of the daily energy requirements (considering a daily requirement of 2000 kcal). Anyway, despite the potential limitations identified in the performance of the device, Turner-McGrievy et al. have shown that the use of the Bite Counter significantly correlates with weight loss [26,27], confirming the potentials of AFIM-BC devices employment in weight management programs.

## 5. Conclusions

Further research is needed to overcome the current limitations of wearable devices in estimating caloric intake, which is not independent of the food being eaten, and to improve the accuracy of the algorithms for the estimation of energy intake through the detection of wrist movement and, more generally, of mouthing activity. As an example, the Italian Ministry of Health (Directorate General for Food Hygiene and Safety and Nutrition) has recently funded the Notion (Measuring caloric intake at population level) research project aimed at developing a unique algorithm capable of estimating energy intake based on mouthing activity recorded by inertial sensors, independent of the type of device used [28]. Undoubtedly, deriving energy intake from mouthing activity is challenging, but itrepresents a promising opportunity from the public health perspective.

## Figures and Tables

**Figure 1 nutrients-11-01170-f001:**
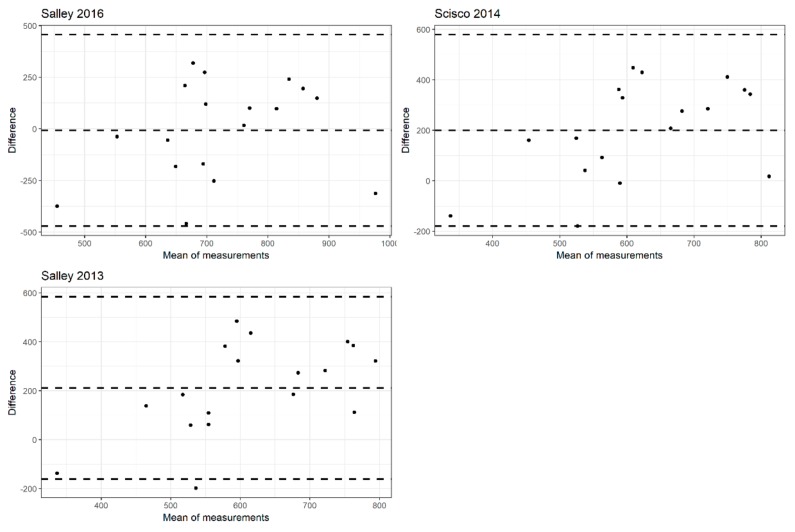
Bland-Altman plots to evaluate the agreement between the actual caloric intake and the estimated caloric intake in Kcal. The means of the measurements are reported on the *x*-axis, whereas the differences between the measurements (McDonald’s nutritional information minus estimated caloric intake) are reported on the *y*-axis.

**Figure 2 nutrients-11-01170-f002:**
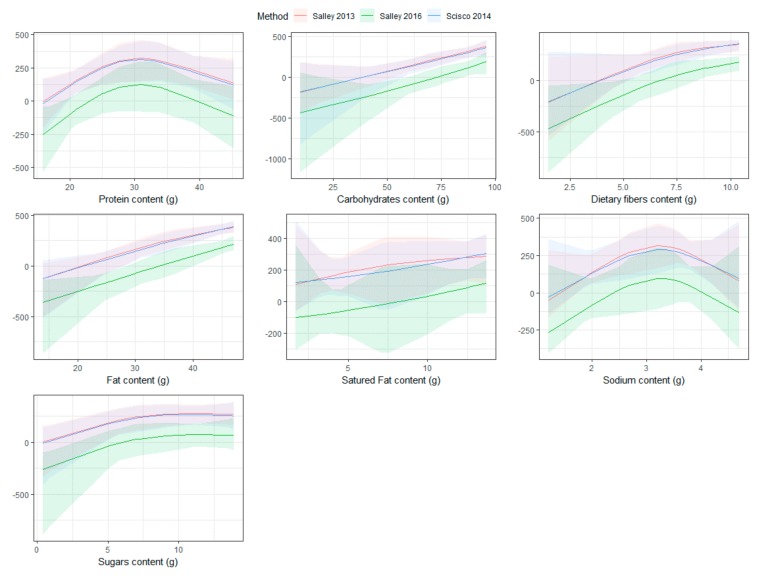
Association between nutritional values in (g), reported on the *x*-axis, and errors in estimated caloric intake in Kcal, reported on the *y*-axis. Lines and bands represent the estimated error and relative 95% CI, respectively. Colors represent the approach used to estimate caloric intake.

**Figure 3 nutrients-11-01170-f003:**
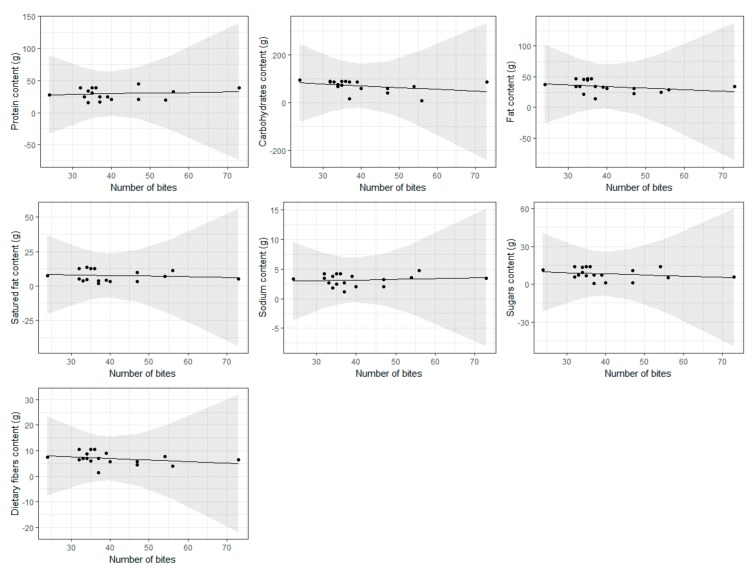
Association between the number of bites, reported on the *x*-axis, and nutritional values in (g), reported on the *y*-axis. Lines and bands represent the estimated error and relative 95% C.I., respectively.

**Table 1 nutrients-11-01170-t001:** Nutritional information for food items included in the study.

	Energy	Protein	Carbs	Fat	Saturated Fat	Sodium	Sugar	Dietary Fiber
CHICKEN McNUGGETS^®^ (6 pieces)	268	17	18	14	1.7	1.2	0.4	1.5
CRISPY McBACON^®^	492	27	32	28	11	1.7	6.3	1.8
McCHICKEN^®^	428	21	46	17	2.3	1.9	6.5	2.8
McWRAP^®^ (grilled chicken)	440	36	37	16	5.5	2.7	8.1	2.1
CHICKEN COUNTRY	479	35	46	17	3.5	2.7	5.3	2.2
HAMBURGER	254	13	30	8.8	3.5	1.3	6.6	2
1955^®^	625	35	49	31	11	2.3	13	4
CHICKENBURGER	344	12	37	16	2.9	1.5	4.4	2.5
McVEGGIE^®^	467	11	65	18	2.8	3.1	11	5.3
QUARTER POUNDER^®^ DELUXE	533	30	35	30	12	1.9	8.4	2.7
CHICKEN WINGS	227	22	3.6	14	3.3	2.5	1.1	0
FRENCH FRIES (Regular)	239	2.7	29	12	1.1	0.55	0.3	2.9
FRENCH FRIES (Medium)	341	3.98	42	17	1.5	0.79	0.5	4.2
VERTIGO FRENCH FRIES (Medium)	330	3.8	41	16	1.7	1.9	0.7	6.1
SALAD	118	9.2	4	6.7	4.3	0.49	2.9	2.4
GREEK SALAD	210	11	6	15	8	2.2	4	4
APPLE SLICES	41	0.2	8.7	0.2	0	0	6.6	2.1

**Table 2 nutrients-11-01170-t002:** Nutritional information for food items included in the study.

ID	Food Item 1	Food Item 2	Food Item 3	Energy (Kcal)	Protein (g)	Carbs (g)	Fat (g)	Saturated Fat (g)	Sodium (g)	Sugar (g)	Dietary Fiber (g)
1	CHICKEN McNUGGETS^®^	FRENCH FRIES (Medium)		609	20.98	60	31	3.2	1.99	0.9	5.7
2	CRISPY McBACON^®^	FRENCH FRIES (Medium)		833	30.98	74	45	12.5	2.49	6.8	6
3	McCHICKEN^®^	FRENCH FRIES (Medium)		769	24.98	88	34	3.8	2.69	7	7
4	McWRAP^®^ (grilled chicken)	SALAD		558	45.2	41	22.7	9.8	3.19	11	4.5
5	CHICKEN COUNTRY	FRENCH FRIES (Medium)		820	38.98	88	34	5	3.49	5.8	6.4
6	HAMBURGER	CHICKENBURGER	FRENCH FRIES (Regular)	837	27.7	96	36.8	7.5	3.35	11.3	7.4
7	CHICKEN COUNTRY	FRENCH FRIES (Medium)		820	38.98	88	34	5	3.49	5.8	6.4
8	1955^®^	VERTIGO FRENCH FRIES (Medium)		955	38.8	90	47	12.7	4.2	13.7	10.4
9	McCHICKEN^®^	FRENCH FRIES (Medium)		769	24.98	88	34	3.8	2.69	7	7
10	1955^®^	VERTIGO FRENCH FRIES (Medium)		955	38.8	90	47	12.7	4.2	13.7	10.4
11	1955^®^	VERTIGO FRENCH FRIES (Medium)		955	38.8	90	47	12.7	4.2	13.7	10.4
12	MENU HAPPY MEAL^®^: HAMBURGER + FRENCH FRIES (Regular) + APPLE SLICES			534	15.9	67.7	21	4.6	1.85	13.5	7
13	CHICKEN McNUGGETS^®^			268	17	18	14	1.7	1.2	0.4	1.5
14	McCHICKEN^®^	VERTIGO FRENCH FRIES (Medium)		758	24.8	87	33	4	3.8	7.2	8.9
15	CHICKEN McNUGGETS^®^	FRENCH FRIES (Medium)		609	20.98	60	31	3.2	1.99	0.9	5.7
16	CHICKEN WINGS	GREEK SALAD		437	33	9.6	29	11.3	4.7	5.1	4
17	McVEGGIE^®^	SALAD		585	20.2	69	24.7	7.1	3.59	13.9	7.7
18	QUARTER POUNDER^®^ DELUXE	VERTIGO FRENCH FRIES (Medium)		863	33.8	76	46	13.7	3.8	9.1	8.8

McNUGGETS^®^ = six pieces.

**Table 3 nutrients-11-01170-t003:** Descriptive statistics of subjects enrolled in the study. Continuous variables are reported as medians [I and III quartiles], and categorical variables are reported as percentages (absolute number).

Subject Characteristics		Overall (*n* = 18)
Age (years)		28.50 [26.00, 29.75]
Gender	Female	61.1 (11)
	Male	38.9 (7)
BMI		21.72 [19.31, 24.83]
Waist		74.50 [68.75, 84.50]
Hip		95.00 [93.00, 95.75]
WHR		0.80 [0.74, 0.90]
Real caloric intake (Kcal)		769.00 [591.00, 836.00]
Protein content (g)		29.34 [21.93, 38.80]
Carbohydrate content (g)		81.50 [61.92, 88.00]
Fat content (g)		34.00 [29.50, 42.95]
Saturated fat content (g)		6.05 [3.85, 12.20]
Sodium content (g)		3.42 [2.54, 3.80]
Sugars content (g)		7.10 [5.80, 12.95]
Dietary fiber content (g)		7.00 [5.78, 8.53]
Number of bites		36.50 [34.00, 45.25]

**Table 4 nutrients-11-01170-t004:** Nutritional content values across percentiles of errors in estimated caloric intake in Kcal for Salley (2013), Scisco (2014), and Salley (2016). The rows where the author’s names are shown report the errors in the estimated caloric intake in Kcal. The results are reported as the median [I and III quartiles]. Differences between median values were assessed using the Kruskal-Wallis test.

Nutritional Value	0–20 Percentile	20–40 Percentile	40–60 Percentile	60–80 Percentile	80–100 Percentile	*p*-Value
Salley (2013)	[−198.2, 80.48]	[80.48, 174.74]	[174.74, 289.95]	[289.95, 383.95]	[383.95, 484.47]	
Protein content (g)	26.60 [19.40, 36.05]	20.98 [18.44, 29.98]	29.39 [23.98, 35.09]	24.98 [24.89, 31.89]	34.89 [30.16, 38.80]	0.84
Carbohydrate content (g)	29.50 [15.90, 48.00]	67.70 [63.85, 77.85]	82.00 [72.00, 88.00]	88.00 [87.50, 89.00]	90.00 [86.00, 91.50]	0.03
Fat content (g)	23.70 [20.52, 25.77]	31.00 [26.00, 32.50]	34.00 [33.25, 37.00]	34.00 [33.50, 40.50]	46.00 [42.95, 47.00]	0.017
Saturated fat content (g)	8.45 [5.75, 10.18]	4.60 [3.90, 4.80]	4.40 [3.65, 7.17]	4.00 [3.90, 8.35]	12.60 [11.25, 12.70]	0.314
Sodium content (g)	3.39 [2.69, 3.87]	1.99 [1.92, 2.74]	3.09 [2.52, 3.57]	3.80 [3.25, 4.00]	3.78 [3.14, 4.20]	0.521
Sugar content (g)	8.05 [3.92, 11.72]	5.80 [3.35, 9.65]	6.40 [4.57, 7.53]	7.20 [7.10, 10.45]	12.50 [10.18, 13.70]	0.447
Dietary fiber content (g)	4.25 [3.38, 5.30]	6.40 [6.05, 6.70]	6.70 [6.23, 7.45]	8.90 [7.95, 9.65]	8.90 [7.05, 10.40]	0.106
Scisco (2014)	[−179, 26.6]	[26.6, 167.2]	[167.2, 293.8]	[293.8, 361.2]	[361.2, 448]	
Protein content (g)	26.60 [19.40, 34.49]	20.98 [18.44, 33.09]	29.39 [23.98, 35.09]	38.80 [31.80, 38.80]	29.34 [27.02, 32.94]	0.905
Carbohydrate content (g)	43.50 [15.90, 73.75]	60.00 [50.50, 63.85]	82.00 [72.00, 88.00]	90.00 [88.50, 90.00]	89.00 [84.50, 91.50]	0.052
Fat content (g)	26.85 [22.02, 30.25]	22.70 [21.85, 26.85]	34.00 [33.25, 37.00]	47.00 [40.00, 47.00]	40.90 [36.10, 45.50]	0.027
Saturated fat content (g)	6.05 [4.17, 8.15]	4.60 [3.90, 7.20]	4.40 [3.65, 7.17]	12.70 [8.35, 12.70]	10.00 [6.58, 12.55]	0.63
Sodium content (g)	3.54 [2.92, 3.87]	1.99 [1.92, 2.59]	3.09 [2.52, 3.57]	4.20 [4.00, 4.20]	3.02 [2.64, 3.56]	0.176
Sugar content (g)	5.45 [3.92, 7.82]	11.00 [5.95, 12.25]	6.40 [4.57, 7.53]	13.70 [10.45, 13.70]	9.15 [6.95, 11.90]	0.368
Dietary fiber content (g)	5.20 [3.38, 6.73]	5.70 [5.10, 6.35]	6.70 [6.23, 7.45]	10.40 [9.65, 10.40]	7.20 [6.75, 8.15]	0.068
Salley (2016)	[−458.67, −224.52]	[−224.52, −40.98]	[−40.98, 103.72]	[103.72, 203.73]	[203.73, 318.56]	
Protein content (g)	26.60 [19.40, 34.49]	20.98 [20.98, 33.09]	29.39 [22.71, 35.09]	38.80 [31.80, 38.80]	29.34 [27.02, 32.94]	0.95
Carbohydrate content (g)	43.50 [15.90, 73.75]	60.00 [50.50, 60.00]	82.00 [73.92, 88.00]	90.00 [88.50, 90.00]	89.00 [84.50, 91.50]	0.04
Fat content (g)	26.85 [22.02, 30.25]	31.00 [26.85, 31.00]	34.00 [30.75, 37.00]	47.00 [40.00, 47.00]	40.90 [36.10, 45.50]	0.057
Saturated fat content (g)	6.05 [4.17, 8.15]	3.20 [3.20, 6.50]	4.80 [4.40, 7.17]	12.70 [8.35, 12.70]	10.00 [6.58, 12.55]	0.495
Sodium content (g)	3.54 [2.92, 3.87]	1.99 [1.99, 2.59]	3.09 [2.48, 3.57]	4.20 [4.00, 4.20]	3.02 [2.64, 3.56]	0.203
Sugar content (g)	5.45 [3.92, 7.82]	0.90 [0.90, 5.95]	8.05 [6.70, 10.20]	13.70 [10.45, 13.70]	9.15 [6.95, 11.90]	0.299
Dietary fiber content (g)	5.20 [3.38, 6.73]	5.70 [5.10, 5.70]	7.00 [6.85, 7.45]	10.40 [9.65, 10.40]	7.20 [6.75, 8.15]	0.032

**Table 5 nutrients-11-01170-t005:** Association between the number of bites and nutritional values. Numbers are reported as the increase in (g) for one more bite (95% C.I.).

Nutritional Value	Increase in (g) for One More Bite	*p*-Value
Protein content (g)	0.0981 (−0.312, 0.508)	0.619
Carbohydrate content	−0.74 (−1.84, 0.364)	0.175
Fat content (g)	−0.253 (−0.683, 0.177)	0.23
Saturated fat content (g)	−0.0433 (−0.235, 0.148)	0.639
Sodium content (g)	0.0124 (−0.0322, 0.0569)	0.565
Sugar content (g)	−0.0889 (−0.298, 0.12)	0.381
Dietary fiber content (g)	−0.0606 (−0.164, 0.0427)	0.231

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
