# Peer review of "Food Composition Impacts the Accuracy of Wearable Devices When Estimating Energy Intake from Energy-Dense Food"

_nutrients, 2019, doi:10.3390/nu11051170_

Round 1

Reviewer 1 Report

Thank you for the opportunity to review this manuscript titled "Food composition impacts the accuracy of wearable devices when estimating energy intake from energy-dense food". The study describes the use of a wrist worn device "Bite Counter" to track estimate the energy intake of 18 subjects when consuming McDonalds. Authors should consider including more references to other literature on technology-based dietary assessment tools, as well as other references where Bite Counter has been used for weight loss, so as to support its potential in public health.

1.     Introduction page 2 – when talking about technologies, it would seem logical to also give reference to the use of mobile apps for dietary assessment, and what limitations there are to these (or recording of food pictures) that would then justify the use of the Bite Counter

2.     Methods, study procedure – could the authors provide some justification as to why McDonalds was chosen as the study setting? Furthermore, if the meals came with drinks, why were beverages excluded from analysis? Technically in real life individuals would consume their food with the beverage (although I understand that by including beverages in analysis, the nutrient/energy-density would be affected).

3.     Results lines 135-139 – please clarify are these median error values, the Bite counter minus the McDonalds nutritional information, or is the difference McDonalds nutritional information minus Bite counter values? It is unclear and confusing.

4.     Table 4 – the percentiles of errors – this should be mentioned in the methods. Also clarification over whether values from Bite counter are minused from McDonalds should be explained (or visa versa). Please add in the caloric differences from the 3 equations into Table 4. I am unsure what the row of values corresponding to the author’s names (e.g. Salley 2013 [-198.2; 80.48]) actually means?

5.     Could the authors consider drawing bland-altman plots to present their findings more clearly? Or in the very least, figure and table legends need to be modified to make it easier to follow as to what they are describing.

6.     I would like the authors to provide more explanation and interpretation of the findings from Figure 1 and Table 5 in their discussion. For example in table 5 – why is there a decrease in grams of all nutrients (except for protein, sodium) with every extra bite? If this is actually the case, then it makes no sense to use this tool since it is not going to be accurate

7.     Discussion – how does the Bite counter compare when eating other foods/snacks (i.e. when it is not a meal) since there may be other hand motions – e.g. eating while working. This should be presented as a limitation, or in the very least consideration for future studies

8.     Discussion – there are 2 studies by Turner-McGrievy 2017 on the Bite Counter and its use in weight loss which the authors should include as key references in their discussion, and particularly with regards to the implications for public health.

9.     Discussion – the fact that only 18 participants were enrolled in the study (and then only of those 15 ate with their hands – to enhance the accuracy of the results), this should be acknowledged as a major limitation to the generalisability of the findings in the discussion, and I don’t think that any conclusions about it’s potential for use in public health is an appropriate conclusion to draw.

Author Response

1.     Introduction page 2 – when talking about technologies, it would seem logical to also give reference to the use of mobile apps for dietary assessment, and what limitations there are to these (or recording of food pictures) that would then justify the use of the Bite Counter

Done

2.     Methods, study procedure – could the authors provide some justification as to why McDonalds was chosen as the study setting? Furthermore, if the meals came with drinks, why were beverages excluded from analysis? Technically in real life individuals would consume their food with the beverage (although I understand that by including beverages in analysis, the nutrient/energy-density would be affected).

Further clarifications have been added to the Study procedure section.

3.     Results lines 135-139 – please clarify are these median error values, the Bite counter minus the McDonalds nutritional information, or is the difference McDonalds nutritional information minus Bite counter values? It is unclear and confusing.

A definition of the error in estimated caloric intake has been added in the Statistical analysis section.

4.     Table 4 – the percentiles of errors – this should be mentioned in the methods. Also clarification over whether values from Bite counter are minused from McDonalds should be explained (or visa versa). Please add in the caloric differences from the 3 equations into Table 4. I am unsure what the row of values corresponding to the author’s names (e.g. Salley 2013 [-198.2; 80.48]) actually means?

A definition of the error in the estimated caloric intake has been added in the Statistical analysis section, i.e. McDonald’s nutritional information minus the estimated caloric intake with one of the evaluated approach. In Table 4, the rows corresponding to the author’s names contain the percentiles values of the error in the estimated caloric intake. For example, the row corresponding to Salley (2013) contains the percentile values of McDonald’s nutritional information minus the caloric intake estimated using the approach from Salley (2013).

5.     Could the authors consider drawing bland-altman plots to present their findings more clearly? Or in the very least, figure and table legends need to be modified to make it easier to follow as to what they are describing.

Bland-Altman plots to evaluate the agreement between actual caloric intake (based on McDonald’s nutritional information) and estimated caloric intake by each approach have been added in Figure 1. The evaluation of the agreement has been added in the Statistical analysis section. A description of the findings on the agreement has been added in the results section.

6.     I would like the authors to provide more explanation and interpretation of the findings from Figure 1 and Table 5 in their discussion. For example in table 5 – why is there a decrease in grams of all nutrients (except for protein, sodium) with every extra bite? If this is actually the case, then it makes no sense to use this tool since it is not going to be accurate

The accuracy of the evaluated approaches in estimating the caloric intake was addressed in Figure 2 (Figure 1 in the original draft), which shows the relationships between nutritional content values and error in estimated caloric intake (McDonald’s nutritional information minus estimated caloric intake based on Bite counter). As explained in the Results section, Salley (2013) and Scisco (2014) methods systematically overestimate the caloric intake, whereas a systematic underestimation was observed for Salley (2016) approach. Moreover, the accuracy of the approaches varies considerably as the nutritional content values vary. These findings suggest a low accuracy for all the evaluated approaches.

One of the goal of the study was to test the eventual correlation between the number of bites and the nutritional content values, as stated in the Statistical analysis section. The results are reported in Table 5, which shows the increase in (g) for each nutritional value for one more bite, along with 95% Confidence Intervals (CIs) and P-values. The same information is also shown graphically in Figure 3 (Figure 2 in the original draft). The results show no statistically significant association between the number of bites and the nutritional content values, as suggested by 95% CIs that include the 0 and P-values higher than 0.05. Thus, those findings did not address the accuracy of the evaluated tools, but the eventual association between the number of bites and nutritional content values. More details about the description of these findings have been added in the results section.

7.     Discussion – how does the Bite counter compare when eating other foods/snacks (i.e. when it is not a meal) since there may be other hand motions – e.g. eating while working. This should be presented as a limitation, or in the very least consideration for future studies

The device was developed to detect bites according to wrist movements. We expect that the wrist movements done for biting something during mealtimes and snack occasions are the same. Differences in the wrist movements patterns have been shown that could depend on the type of utensils used (hand vs utensils, spoon vs. other utensils) and food texture. Consequently, it has been suggested that the validity of the device could be different according to the type of utensils used. Such concept was already included in the first section of the Discussion (“In addition, it has been suggested that the validity of the Bite Counter (in counting the actual number of bites) is different according to food category and that the device is more accurate when subjects eat using their hands [15]”) and as study limitation (Such a limitation could depend on the type of utensils used, which have been shown to influence the wrist movements (e.g., the number of bites required to eat soup with a spoon is often underestimated because the movement of the wrist is reduced to not spill the soup [15]) and on the variability in the size of bite) in the second section of the Discussion.

8.     Discussion – there are 2 studies by Turner-McGrievy 2017 on the Bite Counter and its use in weight loss which the authors should include as key references in their discussion, and particularly with regards to the implications for public health.

Done. The references have been incorporated to the discussion.

9.     Discussion – the fact that only 18 participants were enrolled in the study (and then only of those 15 ate with their hands – to enhance the accuracy of the results), this should be acknowledged as a major limitation to the generalisability of the findings in the discussion, and I don’t think that any conclusions about it’s potential for use in public health is an appropriate conclusion to draw.

Even small, the sample size estimation showed that the number of subjects enrolled in the study was enough to detect an effect size of 0.5 in terms of association between bites and the different nutrients composing the food administered to study participants. Comments about the public health implications for the use of such devices were scaled down.

Reviewer 2 Report

Title: Food composition impacts the accuracy of wearable devices when estimating energy intake from energy dense food.

This is an interesting paper which contributes to the knowledge of the field. However, I have concerns regarding the methodology.

Methods: This section should be improved. It is not exactly explained how the participants were recruited and how energy intake was assessed. Was the food (eg French fries etc) measured ahead and afterwards? How was the exact procedure

Study design: The study design chosen is not really appropriate for the research question. This would be a second step after a well defined laboratory study with a cross-over design (within/between design).

I think the paper needs to address this question if accepted.

Author Response

1)    Methods: This section should be improved. It is not exactly explained how the participants were recruited and how energy intake was assessed. Was the food (eg French fries etc) measured ahead and afterwards? How was the exact procedure

The methods section was improved by providing further details of the study procedures. Energy assessment was based on the nutritional facts reported in the McDonald website for each food item.

2)    Study design: The study design chosen is not really appropriate for the research question. This would be a second step after a well defined laboratory study with a cross-over design (within/between design).I think the paper needs to address this question if accepted.

The validity of the Bite Counter device has been already tested in a controlled laboratory setting (Desendorf et al, 2014). For this reason, we chose to perform the study in a real-life setting, even though subjects were supervised while eating to minimize measure variability.

Round 2

Reviewer 1 Report

The authors have made a good effort with addressing reviewers comments and increasing clarity of the manuscript. A few minor issues to address before acceptance for publication: 

1. Table 4 should still make clear that where the authors names are presented that these rows are representative of caloric differences. Just as the rows below present other nutrients, so this first row should indicate that it’s the different in calories.

2. In the discussion, authors need to actually compare the results of this study with those of Turner-McGrievy and not just merely include these studies as references for another comment

Author Response

1. Table 4 should still make clear that where the authors names are presented that these rows are representative of caloric differences. Just as the rows below present other nutrients, so this first row should indicate that it’s the different in calories.

Further details on the content of Table 4 have been added in the results section and in the caption of Table 4.

2. In the discussion, authors need to actually compare the results of this study with those of Turner-McGrievy and not just merely include these studies as references for another comment

Done